# A VARIANCE PRINCIPLE EXPLAINS WHY DROPOUT FINDS FLATTER MINIMA

## ABSTRACT

Although dropout has achieved great success in deep learning, little is known about how it helps the training find a good generalization solution in the high-dimensional parameter space. In this work, we show that the training with dropout finds the neural network with a flatter minimum compared with standard gradient descent training. We further study the underlying mechanism of why dropout finds flatter minima through experiments. We propose a *Variance Principle* that the variance of a noise is larger at the sharper direction of the loss landscape. Existing works show that SGD satisfies the variance principle, which leads the training to flatter minima. Our work show that the noise induced by the dropout also satisfies the variance principle that explains why dropout finds flatter minima by experiments on various datasets, i.e., CIFAR100, CIFAR10, MNIST and synthetic data, and various structures, i.e., fully-connected networks and large residual convolutional networks. In general, our work points out that the variance principle is an important similarity between dropout and SGD that lead the training to find flatter minima and obtain good generalization.

## 1 INTRODUCTION

Dropout is used with gradient-descent-based algorithms for training DNNs (Hinton et al., 2012; Srivastava et al., 2014). During training, the output of each neuron is multiplied with a random variable with probability $p$ as one and $1-p$ as zero. Note that $p$ is called dropout rate, and every time for computing concerned quantity, the variable is randomly sampled at each feedforward operation. Dropout has been an indispensable trick in the training of deep neural networks (DNNs), however, with very little understanding.

Similar to SGD, training with dropout is equivalent to that with some specific noise. To understand what kind of noise benefits the generalization of training, we proposes a variance principle of a noise, that is,

*Variance Principle:* the variance of a noise is larger at the sharper direction of the loss landscape.

If a noise satisfies the variance principle, it can help the training select flatter minima and leads the training to better generalization. As shown in Zhu et al. (2018); Feng & Tu (2021), the noise in SGD satisfies the variance principle and SGD can find flatter minima and obtain better generalization (Keskar et al., 2016; Neyshabur et al., 2017).

In this work, we study the characteristic of minima learned with dropout. We show that compared with the standard gradient descent (GD), the GD with dropout selects flatter minima. As suggested by many existing works (Keskar et al., 2016; Neyshabur et al., 2017; Zhu et al., 2018), flatter minima are more likely to have better generalization and stability. We then put efforts to show that the noise induced by the dropout satisfies the variance principle, which explains why dropout finds flatter minima.

To examine the variance principle, we explore the relation between the flatness of the loss landscape and the noise structure induced by dropout at minima through three methods and obtain a consistent result that the noise is larger at the sharper direction of the loss landscape to help the training select flatter minima. Our experiments are conducted over synthetic data and fully-connected neural networks as well as modern datasets and models such as, MNIST, CIFAR10 and CIFAR-100 and ResNet-20, thus our conclusion is a rather general result.

First, we examine the inverse variance-flatness relation, similar to Feng & Tu (2021). We define the flatness of a minimum at one direction by the length of the largest interval in the considered direction, which covers the minimum and no point in the interval has loss larger than twice of the loss of the minimum, denoted by $F_{\boldsymbol{p}}$ for the direction $\boldsymbol{p}$. We then consider two definitions of the noise structure, i.e., random trajectory covariance $\Sigma_t$ and random gradient covariance $\Sigma_g$. For the random trajectory covariance, we train the network to an "exploration phase" (Shwartz-Ziv & Tishby, 2017), where the loss decreases with a very slow speed and then sample parameter sets $\{\boldsymbol{\theta}_i\}_{i=1}^N$ from $N$ consecutive training steps to compute the covariance, where $\boldsymbol{\theta}_i$ is the network parameter set at step $i$. For the gradient covariance, we train the network until the loss is very small and then freeze the training. We sample $N$ gradients $\{\boldsymbol{g}_i\}_{i=1}^N$ with different dropout variables to compute the covariance. In each sample, the dropout rate is fixed. For both random trajectory covariance and random gradient covariance, we perform principal component analysis (PCA) and obtain similar results. We find that at the direction of larger variance (larger eigen-value), the loss landscape of the minimum is sharper, i.e., inverse variance-flatness relation.

Second, we study the relation between the Hessian and the noise structure induced by dropout. The eigenvalues of the Hessian of the loss at a minimum are also often used to indicate the flatness. The landscape at an eigen-direction is claimed sharper if the corresponding eigen-value is larger. For each eigen-direction $\boldsymbol{v}_j$, we project the parameter trajectory $\{\boldsymbol{\theta}_i\}_{i=1}^N$ or gradients $\{\boldsymbol{g}_i\}_{i=1}^N$ to the direction of $\boldsymbol{v}_j$ and compute the variance. We find that the noise variance is larger at the direction with a larger eigen-value.

Third, we show that the Hessian matrix aligns well with the random gradient covariance of gradients $\{\boldsymbol{g}_i\}_{i=1}^N$, i.e., their eigen directions of large eigen-values are close, similar to Zhu et al. (2018).

These empirical works show that the noise structure induced by the dropout tends to have larger variance in order to escape the sharper direction, i.e., variance principle, thus, leading to flatter minima. These characteristics of dropout are very similar to SGD (Keskar et al., 2016; Zhu et al., 2018; Feng & Tu, 2021). The similarity between dropout and SGD suggests that modelling their similarity may be a key to understanding how and what stochasticity benefits the training.

## 2 RELATED WORKS

Dropout was proposed as a simple way to prevent neural networks from overfitting, and thus improving the generalization of the network, to a certain extent (Hinton et al., 2012; Srivastava et al., 2014). Many works aim to find an explicit regularization form of dropout. Wager et al. (2013) studies the explicit form of dropout on linear regression and logistic problem, but for studying non-linear neural network, it is still unclear how to characterize the effect of dropout by an explicit regularization term. McAllester (2013) presents PAC-Bayesian bounds, and Wan et al. (2013), Mou et al. (2018) derives Rademacher generalization bounds. These results show that the reduction of complexity brought by dropout is $O(p)$, where $p$ is the probability of keeping an element in dropout. Mianjy & Arora (2020) show that dropout training with logistic loss achieves $\epsilon$-suboptimality in test error in $O(1/\epsilon)$ iterations. All of the above works need specific settings, such as norm assumptions and logistic loss, and they only give a rough estimate of the generalization error bound, which usually consider the worst case. However, it is not clear what is the characteristic of the dropout training process and how to bridge the training with the generalization. In this work, we show that dropout noise has a special structure, which closely relates with the loss landscape. The structure of the effective noise induced by the dropout may be a key reason why dropout can find solutions with better generalization.

Many researches have empirically shown that SGD can improve the generalization performance of neural networks through finding a flatter solution (Li et al., 2017; Jastrzebski et al., 2017; 2018). This work utilizes the current understanding of SGD to study dropout and shows that much similarity is shared between SGD and dropout.

The flatness of the solution is an important aspect of understanding the generalization of neural networks (Keskar et al., 2016; Neyshabur et al., 2017; Zhu et al., 2018). A number of works suggested that the learning rate and batch size determine the flatness of the solutions (Jastrzebski et al., 2017; 2018; Wu et al., 2018). Li et al. (2017) propose a visualization method of the loss landscape at 1-d cross-section to visualize the flatness.

Table 1: Three types of experiments explain why dropout finds flat minima. $\text{Var}(\text{Proj}_{\boldsymbol{v}_i}(S))$ is the variance of the network parameters or gradients projected in the characteristic direction of the Hessian matrix.

| | Dropout covariance $\Sigma$ | |
| --- | --- | --- |
| | Trajectory variance $\Sigma_t$ | gradient variance $\Sigma_g$ |
| Interval flatness $F_{\boldsymbol{v}}$ | $\lambda(\Sigma)$ vs. $F_{\boldsymbol{v}}$, Fig. 2, 3 | |
| Hessian flatness $\lambda(H)$ | $\{\text{Var}(\text{Proj}_{\boldsymbol{v}_i}(S)), \lambda_i(H)\}$, Fig. 4, 5 | |
| | \ | Alignment: $Tr(H\Sigma_g)$, Fig. 6. |

Feng & Tu (2021) investigate the connection between SGD learning dynamics and the loss landscape through the principal component analysis (PCA), and show that SGD dynamics follow a low-dimensional drift-diffusion motion in the weight space. Through characterizing the loss landscape by its flatness in each PCA direction around the solution found by SGD, they also reveal a robust inverse relation between the weight variance and the landscape flatness in PCA directions, thus finding that SGD serves as a landscape dependent annealing algorithm to search for flat minima.

Zhu et al. (2018) study a general form of gradient based optimization dynamics with unbiased noise to analyze the behavior of SGD on escaping from minima and its regularization effects. They also introduce an indicator to characterize the efficiency of escaping from minima through measuring the alignment of noise covariance and the curvature of loss function and thus revealing the anisotropic noise of SGD.

## 3 PRELIMINARY

### 3.1 DROPOUT

Consider an $L$-layer neural network $f_{\boldsymbol{\theta}}(\boldsymbol{x})$. With dropout (Srivastava et al., 2014), the feedforward operation in a network $f_{\boldsymbol{\theta}}(\boldsymbol{x})$ is

$$f_{\boldsymbol{\theta}}^{[0]}(\boldsymbol{x}) = \boldsymbol{x}, \tag{1}$$

$$r_j \sim \text{Bernoulli}(p), \tag{2}$$

$$f_{\boldsymbol{\theta}}^{[l]}(\boldsymbol{x}) = \boldsymbol{r}^{[l]} \circ \sigma \circ (\boldsymbol{W}^{[l-1]} f_{\boldsymbol{\theta}}^{[l-1]}(\boldsymbol{x}) + \boldsymbol{b}^{[l-1]}) \quad 1 \leq l \leq L-1, \tag{3}$$

$$f_{\boldsymbol{\theta}}(\boldsymbol{x}) = f_{\boldsymbol{\theta}}^{[L]}(\boldsymbol{x}) = \boldsymbol{W}^{[L-1]} f_{\boldsymbol{\theta}}^{[L-1]}(\boldsymbol{x}) + \boldsymbol{b}^{[L-1]}, \tag{4}$$

where $p$ is the dropout rate, $\boldsymbol{W}^{[l]} \in \mathbb{R}^{m_{l+1} \times m_l}$, $\boldsymbol{b}^{[l]} = \mathbb{R}^{m_{l+1}}$, $m_0 = d_{\text{in}} = d$, $m_L = d_{\text{o}}$, $\sigma$ is a scalar function and "$\circ$" means entry-wise operation. We denote the set of parameters by

$$\boldsymbol{\theta} = (\boldsymbol{W}^{[0]}, \boldsymbol{W}^{[1]}, \ldots, \boldsymbol{W}^{[L-1]}, \boldsymbol{b}^{[0]}, \boldsymbol{b}^{[1]}, \ldots, \boldsymbol{b}^{[L-1]}),$$

### 3.2 INTERVAL FLATNESS

We use the definition of flatness in Feng & Tu (2021). For convenience, we call it *interval flatness* Around a specific solution $\boldsymbol{\theta}_0^*$, we compute the loss function profile $L_{\boldsymbol{v}}$ along the direction $\boldsymbol{v}$:

$$L_{\boldsymbol{v}}(\delta\theta) \equiv L(\boldsymbol{\theta}_0^* + \delta\theta\boldsymbol{v}).$$

The interval flatness $F_{\boldsymbol{v}}$ is defined as the width of the region within which $L_{\boldsymbol{v}}(\delta\theta) \leq 2L_{\boldsymbol{v}}(0)$. We determine $F_{\boldsymbol{v}}$ by finding two closest points $\boldsymbol{\theta}_{\boldsymbol{v}}^l < 0$ and $\boldsymbol{\theta}_{\boldsymbol{v}}^r > 0$ on each side of the minimum that satisfy $L_{\boldsymbol{v}}(\boldsymbol{\theta}_{\boldsymbol{v}}^l) = L_{\boldsymbol{v}}(\boldsymbol{\theta}_{\boldsymbol{v}}^r) = 2L_{\boldsymbol{v}}(0)$. The scale factor 2 is used in Feng & Tu (2021), and after our test, the result is not sensitive to the selection of this factor. In this work, we follow their experimental scheme to show the similarity between dropout and SGD. The interval flatness is defined as:

$$F_{\boldsymbol{v}} \equiv \boldsymbol{\theta}_{\boldsymbol{v}}^r - \boldsymbol{\theta}_{\boldsymbol{v}}^l. \tag{5}$$

A larger value of $F_{\boldsymbol{v}}$ means a flatter landscape in the direction $\boldsymbol{v}$.

### 3.3 RANDOMNESS INDUCED BY DROPOUT

#### 3.3.1 RANDOM TRAJECTORY DATA

The training process of neural networks are usually divided into two phases, fast convergence and exploration phase (Shwartz-Ziv & Tishby, 2017). Feng & Tu (2021)'s work focuses on the behavior of networks in the exploration phase. In this work, we follow the experimental scheme in Feng & Tu (2021) to show the similarity between dropout and SGD. This can be understood by frequency principle (Xu et al., 2019; 2020), which states that DNNs fast learn low-frequency components but slowly learn high-frequency ones.

We collect parameter sets $S_{para} = \{\boldsymbol{\theta}_i\}_{i=1}^N$ from $N$ consecutive training steps in the exploration phase, where $\boldsymbol{\theta}_i$ is the network parameter set at step $i$.

#### 3.3.2 RANDOM GRADIENT DATA

However, due to the limitation of the number of sampling points, we often need larger time interval for sampling. Although the network loss is small, compared with the initial sampling parameters, the network parameters could have large changes during the long-time sampling. Therefore, much extra noise may be induced. Meanwhile, for dropout, it is difficult to get a small loss value on large networks and datasets, therefore, model parameters often have large fluctuations. Based on this problem, we propose a more appropriate sampling method to avoid additional noise caused by sampling parameters in a large time interval. We train the network until the loss is very small and then freeze the training. We sample $N$ gradients of the loss function w.r.t. the parameters with different dropout variables, i.e., $S_{grad} = \{\boldsymbol{g}_i\}_{i=1}^N$. In each sample, the dropout rate is fixed. In this way, we can get the noise structure of dropout without being affected by parameter changes caused by long-term training.

### 3.4 INVERSE VARIANCE-FLATNESS RELATION

We study the inverse variance flatness relation for both random trajectory data and random gradient data. For convenience, we denote data as $S$ and its covariance as $\Sigma$.

#### 3.4.1 INTERVAL FLATNESS VS. VARIANCE

Following Feng & Tu (2021), we perform principal component analysis (PCA) for $\Sigma$. Denote $\lambda_i(\Sigma)$ as the $i$-th eigenvalue of $\Sigma$, and denote its corresponding eigen-vector as $\boldsymbol{v}_i(\Sigma)$. We then compute the flatness of the loss landscape in the direction $\boldsymbol{v}_i(\Sigma)$, and denote it as $F_{\boldsymbol{v}_i(\Sigma)}$. Finally, we display the scatter plot of $\{\lambda_i(\Sigma), F_{\boldsymbol{v}_i(\Sigma)}\}$.

#### 3.4.2 PROJECTED VARIANCE VS. HESSIAN FLATNESS

The method in Feng & Tu (2021), requires many training steps to obtain the covariance. To obtain the variance induced by the dropout at a fixed position $\boldsymbol{\theta}$, we propose another way to characterize the inverse variance-flatness relation. We use the the eigen-value $\lambda_i(H)$ of the Hessian $H$ of the loss landscape at $\boldsymbol{\theta}$ to denote the sharpness. Hessian matrix is the matrix obtained by the second derivative of the loss function of the neural network with respect to the parameter vector of neural network. Here, the parameter vector is a vector consisting of all the parameters of network. The variance induced by dropout at the corresponding eigen-direction $\boldsymbol{v}_i(H)$ is computed by the following procedure. For each eigen-direction $\boldsymbol{v}_i$ of Hessian $H$, we project the sampled parameters or the gradients of sampled parameter $S$ to direction $\boldsymbol{v}_i$ by inner product, denoted by $\mathrm{Proj}_{\boldsymbol{v}_i}(S)$. Then, the variance at direction $\boldsymbol{v}_i(H)$ is the variance of dataset $\mathrm{Proj}_{\boldsymbol{v}_i}(S)$, denoted by $\mathrm{Var}(\mathrm{Proj}_{\boldsymbol{v}_i}(S))$. Finally, we display the scatter plot of $\{\mathrm{Var}(\mathrm{Proj}_{\boldsymbol{v}_i}(S)), \lambda_i(H)\}$.

### 3.5 ALIGNMENT BETWEEN HESSIAN AND GRADIENT COVARIANCE

We use the method in Zhu et al. (2018) to quantify the alignment between the noise structure and the curvature of loss surface. For each training step $i$, we calculate the alignment parameter $T_i$:

$$T_i = \mathrm{Tr}(H_i \Sigma_i),$$

where $\Sigma_i$ is the $i$th-step covariance matrix generated by dropout layers and $H_i$ is the $i$th-step Hessian matrix of network parameters.

## 4 EXPERIMENTAL SETUP

To understand the effects of dropout, we trained a number of networks with different structures. We consider the following types of neural networks: 1) Fully-connected neural networks (FNNs) trained by MNIST. For FNN, all parameters are initialized by Xavier initialization(Glorot & Bengio, 2010). 2) Convolutional neural networks (CNNs) trained by CIFAR-10. For CNN, all parameters are initialized by He initialization (He et al., 2015). The loss of all our experiments is cross entropy loss. 3) Deep residual neural networks (ResNets) (He et al., 2016) trained by CIFAR-100. For ResNet, all parameters are initialized by He initialization (He et al., 2015). The loss of all our experiments is cross entropy loss.

For Fig. 1 (a), we use the FNN with size $784 - 1024 - 1024 - 10$. We add dropout layers behind the first and the second layers with dropout rate of 0.8 and 0.5, respectively. We train the network using default Adam optimizer (Kingma & Ba, 2015) with a learning rate of 0.0001.

For Fig. 1 (b), we use vgg-9 (Simonyan & Zisserman, 2014) to compare the loss landscape flatness w/o dropout layers. Models are trained using GD with Nesterov momentum, training-size 2048 for 300 epochs. The learning rate was initialized at 0.1, and divided by a factor of 10 at epochs 150, 225 and 275. We only use 2048 examples for training to compromise with the computational burden.

For Fig. 1 (c), 3, 5, we use ResNet-20 (He et al., 2016) to compare the loss landscape flatness w/o dropout layers. Models are trained using GD, training-size 50000 for 1200 epochs. The learning rate was initialized at 0.01. Since the Hessian calculation of ResNet takes much time, for the ResNet experiment, we only perform it at a specific dropout rate and learning rate.

For Fig. 2, 4, 6, we use the FNN with size $784 - 50 - 50 - 10$. We train the network using GD with the first 10,000 training data as the training set. We add a dropout layer behind the second layer. The dropout rate and learning rate are specified and unchanged in each experiment.

It is worth noting that, in order to avoid the influence of the noise induced by SGD in our experiments, all our networks are trained using GD, so it is difficult for us to verify on larger datasets such as ImageNet.

## 5 DROPOUT FINDS FLATTER MINIMA

Dropout is almost ubiquitous in training deep networks. It is interesting and important to understand what makes dropout improve the generalization of training neural networks. Inspired by the study of SGD (Keskar et al., 2016), we explore the flatness of the minima found by dropout.

The loss landscape of DNNs is highly non-convex and complicate (Skorokhodov & Burtsev, 2019), but with certain characteristics, such as embedding principle (Zhang et al., 2021) shows that the loss landscape of any network "contains" all critical points of all narrower networks, in the sense that, any critical point of any narrower network can be embedded to a critical point of the target network preserving its output function. It is impractical to visualize the loss landscape in the high-dimensional space. To compare two minima, $\boldsymbol{\theta}$ and $\boldsymbol{\theta}'$, in a loss landscape, one simple way (Keskar et al., 2016) is to visualize 1-d cross-section of the interpolation between $\boldsymbol{\theta}$ and $\boldsymbol{\theta}'$. However, empirical studies (Li et al., 2017) show that this simple way could be misleading when $\boldsymbol{\theta}$ and $\boldsymbol{\theta}'$ have difference of orders of magnitude. Li et al. (2017) visualizes loss functions using filter-wise normalized directions to remove the scaling effect. We adopt this method (Li et al., 2017) in this work as follows. To obtain a direction for a network with parameters $\boldsymbol{\theta}$, we begin by producing a random Gaussian direction vector $\boldsymbol{d}$ with dimensions compatible with $\boldsymbol{\theta}$. Then, we normalize each filter in $\boldsymbol{d}$ to have the same norm of the corresponding filter in $\boldsymbol{\theta}$. In other words, we make the replacement $\boldsymbol{d}_{i,j} \leftarrow \frac{\boldsymbol{d}_{i,j}}{\|\boldsymbol{d}_{i,j}\|} \|\boldsymbol{\theta}_{i,j}\|$, where $\boldsymbol{d}_{i,j}$ represents the $j$th filter (not the $j$th weight, for FNN, one layer is one filter.) of the $i$th layer of $\boldsymbol{d}$, and $\|\cdot\|$ denotes the Frobenius norm. We use $f(\alpha) = L(\boldsymbol{\theta} + \alpha \boldsymbol{d})$ to characterize the loss landscape around the minima obtained with dropout layers $\boldsymbol{\theta}_{\boldsymbol{d}}^*$ and without dropout layer $\boldsymbol{\theta}^*$. For all network structures shown in Fig. 1, dropout can improve the generalization of the network and find a flatter minimum. In Fig. 1(a), (b), for both networks trained with and

without dropout layers, the training loss values are all closed to zero, but their flatness and generalization are still quite different. In Fig. 1(c), due to the complexity of the dataset and network, and the large number of dropout layers, the loss value of network with dropout layers is larger than the one without dropout layer. However, the network with dropout layers still finds a flatter minimum with better generalization.

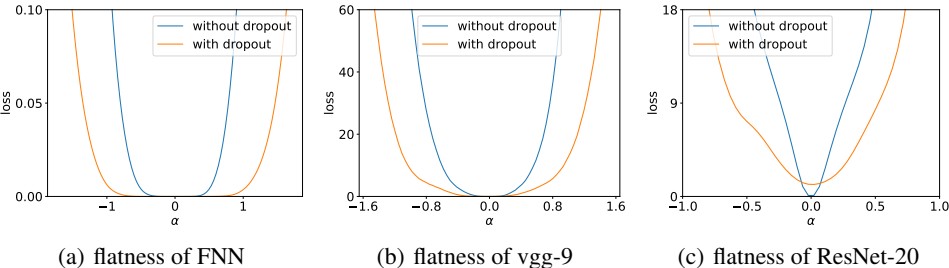

(a) flatness of FNN          (b) flatness of vgg-9          (c) flatness of ResNet-20

Figure 1: The 1D visualization of solutions of different network structures obtained with or without dropout layers. (a) The FNN is trained on MNIST dataset. For experiment with dropout layers, we add dropout layer after the first and the second layers, the dropout rates of the two dropout layers are 0.8 and 0.5, respectively. The test accuracy for model with dropout layers is 98.7% while 98.1% for model without dropout layers. (b) The vgg-9 network is trained on CIFAR-10 dataset using the first 2048 examples as training dataset. For experiment with dropout layers, we add dropout layers after the pooling layers, the dropout rates of dropout layers are 0.8. The test accuracy for model with dropout layers is 60.6% while 59.2% for model without dropout layers. (c) The ResNet-20 network is trained on CIFAR-100 dataset using the 50000 examples as training dataset. For experiment with dropout layers, we add dropout layers after the convolutional layers, the dropout rates of dropout layers are 0.8. The test accuracy for model with dropout layers is 54.7% while 34.1% for model without dropout layers.

## 6 INVERSE VARIANCE-FLATNESS RELATION

Similar to SGD, the effect of dropout can be equivalent to imposing a specific noise on the gradient. A random noise, such as isotropic noise, can help the training escape local minima, but can not robustly improve generalization (An, 1996; Zhu et al., 2018). The noise induced by the dropout should have certain properties that can lead the training to good minima.

In this section, we show that the noise induced by the dropout satisfies the variance principle, that is, the noise variance is larger along the sharper direction of the loss landscape at a minimum. The landscape-dependent structure helps the training escape sharp minima. We utilize three methods to examine the variance principle for dropout, as summarized in Table 1.

### 6.1 INTERVAL FLATNESS VS. VARIANCE

We use the principal component analysis (PCA) to study the weight variations when the accuracy is nearly 100%. For FNNs, networks are trained on MNIST with the first 10000 examples as the training set for computational efficiency. For ResNets, networks are trained on CIFAR-100 with 50000 examples as the training set to make a more convincing conclusion on modern datasets and models. The networks are trained with full batch for different learning rates and dropout rates under the same random seed (that is, with the same initialization parameters). After the loss is small enough, we sample the parameters or gradients of parameters $N$ times ($N = 3000$ in this experiment) and use the method introduced in Sec. 3.3 to construct covariance matrix $\Sigma$ by the weights $S_{para}$ or gradients $S_{grad}$ between two hidden layers. The PCA was done for the covariance matrix $\Sigma$. We then compute the interval flatness of the loss function landscape at eigen-directions, i.e., $\{F_{\boldsymbol{v}_i(\Sigma)}\}_{i=1}^{N}$. Note that the PCA spectrum $\{\lambda_i(\Sigma)\}_{i=1}^{N}$ indicate the variance of weights $S_{para}$ or gradients $S_{grad}$ at corresponding eigen-directions.

As shown in Fig. 2, 3, for different learning rates and dropout rates, there is an inverse relationship between the interval flatness of the loss function landscape $\{F_{\boldsymbol{v}_i(\Sigma)}\}_{i=1}^{N}$ and the dropout variance,

i.e., the PCA spectrum $\{\lambda_i(\Sigma)\}_{i=1}^N$. We can approximately see a power-law relationship between $\{F_{\boldsymbol{v}_i(\Sigma)}\}_{i=1}^N$ and $\{\lambda_i(\Sigma)\}_{i=1}^N$ for different dropout rates and learning rates. More detailed, for small flatness part, the variance of noise induced by dropout is generally large, which indicates that the noise induced by dropout has larger variance in sharp directions, for large flatness part, as the loss landscape flatter, the linear relationship more obvious, we can see a clearer asymptotic behavior in the results. Overall, we can observe the negative correlation between the eigenvalues and flatness in Fig. 2, 3.

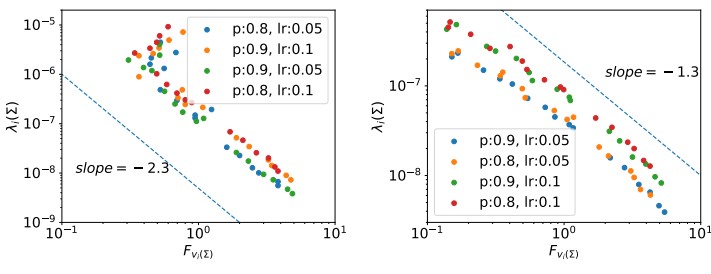

(a) datasets $S$ sampled from parame-  (b) datasets $S$ sampled from gradi-
ters                                     ents of parameters

Figure 2: The inverse relation between the variance $\{\lambda_i(\Sigma)\}_{i=1}^N$ and the flatness $\{F_{\boldsymbol{v}_i(\Sigma)}\}_{i=1}^N$ for different choices of dropout rate $p$ and learning rate $lr$. The FNN is trained on MNIST dataset using the first 10000 examples as training dataset. The PCA is done for different datasets $S$ sampled from parameters for (a) and sampled from gradients of parameters for (b). The dash lines give the approximate slope of the scatter.

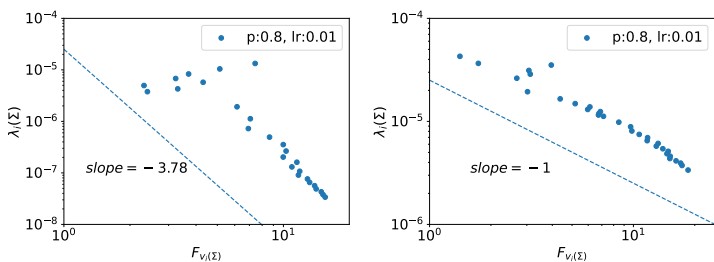

(a) datasets $S$ sampled from parame-  (b) datasets $S$ sampled from gradi-
ters                                     ents of parameters

Figure 3: The inverse relation between the variance $\{\lambda_i(\Sigma)\}_{i=1}^N$ and the flatness $\{F_{\boldsymbol{v}_i(\Sigma)}\}_{i=1}^N$ for different choices of dropout rate $p$ and learning rate $lr$. The ResNet is trained on CIFAR-100 dataset. The PCA is done for different datasets $S$ sampled from parameters for (a) and sampled from gradients of parameters for (b). The dash lines give the approximate slope of the scatter.

## 6.2 PROJECTED VARIANCE VS. HESSIAN FLATNESS

The eigenvalues of the Hessian of the loss at a minimum are also often used to indicate the flatness. A large eigenvalue correspond to a sharper direction. In this section, we study the relationship between eigen-values of the Hessian $H$ of parameters $\boldsymbol{\theta}$ at the end point of training and the variances of dropout at corresponding eigen-directions. As mentioned in the Preliminary section, we sample the parameters or gradients of parameters 1000 times, that is $N = 1000$. For each eigen-direction $\boldsymbol{v}_i$ of Hessian $H$, we project the sampled parameters or the gradients of sampled parameter to direction $\boldsymbol{v}_i$ by inner product, denoted by $\text{Proj}_{\boldsymbol{v}_i}(S)$. Then, we compute the variance of the projected data, i.e., $\text{Var}(\text{Proj}_{\boldsymbol{v}_i}(S))$.

As shown in Fig. 4, 5, we find that there is also a power-law relationship between $\{\lambda_i(H)\}_{i=1}^D$ and $\{\text{Var}(\text{Proj}_{\boldsymbol{v}_i}(S))\}_{i=1}^D$ for different dropout rates and learning rates, no matter if $S$ is sampled

from parameters or gradients of parameters. The positive correlation between the eigenvalue and the projection variance show the structure of the dropout noise, which helps the network escape the bad minima. At the same time, as shown in Fig. 2, 3, 4, 5, we can see that gradient sampling has a more obvious linear structure than parameter sampling, which shows that gradient sampling can better avoid the interference of irrelevant noise.

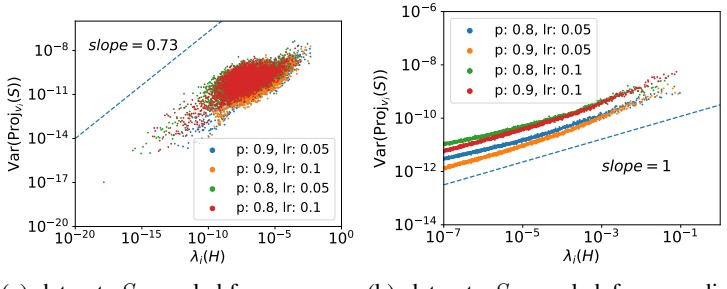

(a) datasets $S$ sampled from parameters  (b) datasets $S$ sampled from gradients of parameters

Figure 4: The relation between the variance $\{\mathrm{Var}(\mathrm{Proj}_{\boldsymbol{v}_i}(S))\}_{i=1}^D$ and the eigenvalue $\{\lambda_i(H)\}_{i=1}^D$ for different choices of dropout rate $p$ and learning rate $lr$. The FNN is trained on MNIST dataset using the first 10000 examples as training dataset. The projection is done for different datasets $S$ sampled from parameters for (a) and sampled from gradients of parameters for (b). The dash lines give the approximate slope of the scatter.

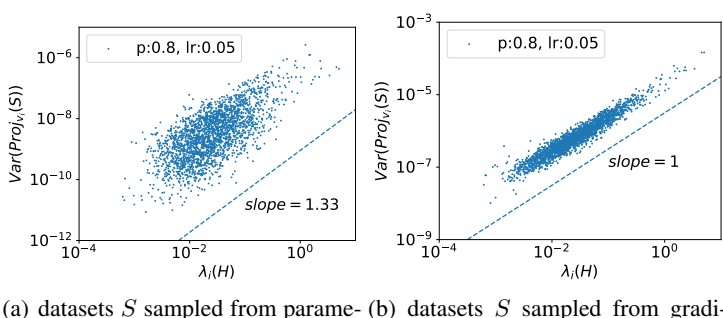

(a) datasets $S$ sampled from parameters  (b) datasets $S$ sampled from gradients of parameters

Figure 5: The relation between the variance $\{\mathrm{Var}(\mathrm{Proj}_{\boldsymbol{v}_i}(S))\}_{i=1}^D$ and the eigenvalue $\{\lambda_i(H)\}_{i=1}^D$ for different choices of dropout rate $p$ and learning rate $lr$. The ResNet is trained on CIFAR-100 dataset using all the examples as training dataset. The projection is done for different datasets $S$ sampled from parameters for (a) and sampled from gradients of parameters for (b). The dash lines give the approximate slope of the scatter.

## 6.3 ALIGNMENT BETWEEN HESSIAN AND GRADIENT COVARIANCE

Zhu et al. (2018) show that the alignment indicator $\mathrm{Tr}(H\Sigma)$ plays an crucial role for stochastic processes escaping from minima, where $H$ is the Hessian and $\Sigma$ is the noise covariance. In this subsection, we study the alignment between the Hessian and the random gradient covariance at each training step. Note that the training is performed by GD without dropout. At step $i$, we sample the gradients of parameters $\{\boldsymbol{g}_i^j\}_{j=1}^N$ by tentatively adding a dropout layer between the hidden layers. For each step $i$, we the compute $\mathrm{Tr}(H_i\Sigma_i)$, where $H_i$ is the Hessian of the loss at the parameter set at step $i$ and $\Sigma_i$ is the covariance of $\{\boldsymbol{g}_i^j\}_{j=1}^N$.

In order to show the anisotropic structure, we construct the isotropic noise for comparison, i.e., $\bar{\Sigma}_i = \frac{\mathrm{Tr}\,\Sigma_i}{D}I$ of the covariance matrix $\Sigma_i$, where $D$ is the number of parameters. In our experiments, $D = 2500$. As shown in Fig. 6 , in the whole training process under different learning rates

and dropout rates, $\mathrm{Tr}(H_i\Sigma_i)$ is much larger than $\mathrm{Tr}(H_i\bar{\Sigma}_i)$, indicating the anisotropic structure of dropout noise and its high alignment with the Hessian matrix.

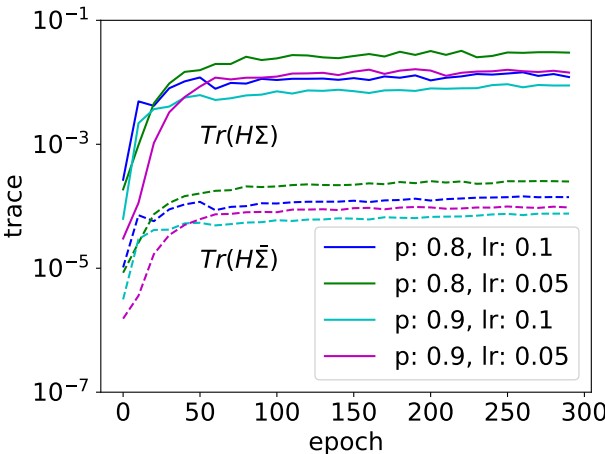

Figure 6: Comparison between $\mathrm{Tr}(H_i\Sigma_i)$ and $\mathrm{Tr}(H_i\bar{\Sigma}_i)$ in each training epoch $i$ for different choices of dropout rate $p$ and learning rate $lr$. The FNN is trained on MNIST dataset using the first 10000 examples as training dataset. The solid and the dotted lines represent the value of $\mathrm{Tr}(H_i\Sigma_i)$ and $\mathrm{Tr}(H_i\bar{\Sigma}_i)$, respectively.

## 7 CONCLUSION AND DISCUSSION

In this work, we propose a variance principle that noise is larger at the sharper direction of the loss landscape. If a noise satisfies the variance principle, it helps the training select flatter minima and leads the training to good generalization. We empirical show that dropout finds flat minima during the training and examine that the dropout satisfies the variance principle to explain why dropout finds flat minima over various datasets and neural network structures by the following three perspectives: interval flatness vs. variance, projected variance vs. Hessian flatness and alignment between Hessian and gradient covariance.

The dropout and the SGD are common in sharing the variance principle. Based on this, there could be a general theory framework that can model both dropout and SGD to understand why they have better generalization. The research on understanding the good generalization of dropout and SGD is far from complete. As a starting point, the variance principle shows a promising and reasonable direction for understanding the stochastic training of neural networks.

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
