# OpenReview forum: "A Variance Principle Explains why Dropout Finds Flatter Minima"
_ICLR.cc/2022/Conference — ICLR 2022 Submitted_

### Official Review · Reviewer_Y6pQ · 2021-10-24

**Correctness:** 3
**Technical Novelty And Significance:** 2
**Empirical Novelty And Significance:** 3
**Recommendation:** 5
**Confidence:** 5

**Main Review:**


To my best knowledge, this is a first work that shows the relationship between Dropout and the flatness of minima. It could provide another perspective for understanding the effectiveness of Dropout.

However, my main concern about this manuscript is the complete missing of theoretical analysis, which makes the current version of the paper not strong enough for publication. I strongly suggest the authors conduct theoretical analysis over the covariance introduced by Dropout noise and loss curvature, at least on linear cases.


**Summary Of The Paper:**

This paper empirically found that Dropout noise could help the neural network search for the flat minima, by showing that the covariance of parameter and gradient align well with the loss curvature and have inverse relationship with interval flatness.

**Summary Of The Review:**

Complete missing of theoretical analysis.

---

> ### Author Response · Authors · 2021-11-16
> **Response to Reviewer Y6pQ**
>
> $\textbf{Point 1:}$
> Main concern about this manuscript is the complete missing of theoretical analysis, which makes the current version of the paper not strong enough for publication.
>
> $\textbf{Reply:}$
>
> Dear reviewer, we would like to argue that this should not be a concern.
> Although theoretical analysis is an important part of deep learning, theoretical analysis should not be a mandatory choice for  the basic research. This paper is an empirical study on the basic research of deep learning, while the theoretical part is left for further study. Many important papers have obtained important results only through experiments, which has a significant impact on the research of deep learning, for example, the best paper of ICLR: Zhang et al. (2017), Understanding deep learning requires rethinking generalization.

---

> ### Author Response · Authors · 2021-11-22
> **Large resnet and CIFAR100 results added**
>
> We have updated the draft with new experiments on resnet20 and CIFAR100. There are three methods in our paper to verify the variance principle. We have done experiments on the first and the second method and obtain similar results in such complicated dataset and network structure. Note that we do not use the third method in the resnet20, because this method requires computing Hessian while computation cost is too large.

---

> ### Author Response · Authors · 2021-12-09
> **Transformer and Multi30k results added**
>
> Dear reviewers,
>
> We have updated the draft with new experiments on transformer and Multi30k dataset.  We have done experiments on the first and the second method and obtain similar results in the new experimental setup. Up to now, our experiments are conducted over several representative datasets, i.e., MNIST, CIFAR-10, CIFAR-100 and Multi30k, and network structures, i.e., fully-connected neural networks, VGG-9. ResNet-20 and transformer, thus our conclusion is a rather general result. Since we cannot update the draft on openreview, we post the new version on the anonymous website (details can be found on https://www.dropbox.com/s/5kd7q7wevi984yl/Why_dropout_finds_flatter_minima.pdf?dl=0).  Based on a large number of more complex experiments, we expect reviewers to raise the score appropriately.
>
> Sincerely yours,
>
> Authors.

---

### Official Review · Reviewer_dtVh · 2021-10-30

**Correctness:** 2
**Technical Novelty And Significance:** 2
**Empirical Novelty And Significance:** 2
**Recommendation:** 5
**Confidence:** 2

**Main Review:**

My detailed comments are given as below.

Strength:
1. This paper is well written and the motivation is clear to me. Dropout is a widely used technique for improving the generalization performance of various learning algorithms, but there is limited understanding how this scheme helps to achieve this better generalization. This paper provides some empirical justification showing that dropout can lead to flatter minima.

Weakness:
1. Some parts need to be explained more clearly. For example, I have a hard time understanding Section 3.3., where interval flatness is defined as the width of the region where $L_v(\delta \theta)<2L_v(0)$. I am wondering why $2L_v(0)$ is chosen here, and whether the factor $2$ is sufficiently general to characterize the landscapes for all settings. I am not familiar with this part, but I hope the authors should explain it a little bit.

2. My second concern is the novelty of the results. So far I see that some important tools, e.g., flatness definition, PCA, follow from existing studies, although there is some new treatment in obtain the variance. In addition, the empirical results are not that surprising to me (but I agree that it is not clear why the noise induced by dropout plays some similar role as SGD), and a more interesting part may be to provide some theoretical justification even in some simplified settings.

3. Experiments can be strengthened. I see that all experiments are conducted over small datasets (e.g., CIFAR-10) and small models. To make a more convincing conclusion that dropout can lead to flatter minima, I believe experiments on modern datasets and models (e.g., ResNet on ImageNet) need to be run.







**Summary Of The Paper:**

This paper provides an empirical study on how the dropout can lead to minima within a flatter landscape (hence a better generalization performance). Based on the definition of minimizer flatness (i.e., eq. 5) and random trajectory data (parameters from a optimization path and gradients when loss is stably small), the authors first show that dropout can help to find flatter minima, and also find a inverse relation between the flatness (the length $F_v$) and the algorithmic variance (i.e., in area with sharper direction, algorithm with dropout has a larger variance), and further find that for areas with larger eigenvalues (i.e., with sharper landscape), the variances of the algorithm along the directions of their eigenvectors are also large. Such empirical results together demonstrate that the dropout can play a good role in improving the generalization of existing algorithms by introducing certain level of noise.

**Summary Of The Review:**

I think this paper provides some empirical explanation on why dropout is useful to achieve a flatter minima. However,  I feel it is more interesting to rigorously explain this phenomenon given that many works already prove that SGD can reach flatter minima. In addition, the novelty of this paper is not that high based on my understanding. Therefore, I am slightly negative about this paper, but I am open to adjust my score based on the authors' feedback and other reviewers' comments.

---

> ### Author Response · Authors · 2021-11-16
> **Response to Reviewer dtVh**
>
> $\textbf{Point 1:}$
>  Why $2L_v(0)$ is chosen here, and whether the factor is sufficiently general to characterize the landscapes for all settings.
>
> $\textbf{Reply:}$
>
> Feng \& Tu (2021) use a scale factor of 2 as an example to define the flatness. They mention in the paper that the size of this scale factor has little effect on flatness. Meanwhile, we also find that the relationship between flatness in different directions is not sensitive to the selection of this factor. In this work, we follow their experimental scheme to show the similarity between dropout and SGD. For clarification, we add a description in the paper:
>
> "The scale factor 2 is used in Feng \& Tu (2021), and after our test, the result is not sensitive to the selection of this factor. In this work, we follow their experimental scheme to show the similarity between dropout and SGD."
>
> $\textbf{Point 2:}$
> A more interesting part may be to provide some theoretical justification even in some simplified settings.
>
> $\textbf{Reply:}$
>
> Dear reviewer, we would like to argue that this should not be a concern.
> Although theoretical analysis is an important part of deep learning, theoretical analysis should not be a mandatory choice for  the basic research. This paper is an empirical study on the basic research of deep learning, while the theoretical part is left for further study. Many important papers have obtained important results only through experiments, which has a significant impact on the research of deep learning, for example, the best paper of ICLR: Zhang et al. (2017), Understanding deep learning requires rethinking generalization.
>
> $\textbf{Point 3:}$
> To make a more convincing conclusion that dropout can lead to flatter minima, I believe experiments on modern datasets and models (e.g., ResNet on ImageNet) need to be run.
>
> $\textbf{Reply:}$
>
> Dear reviewer, we would like to argue that this should also not be a concern.
> It is difficult to require researchers in university to perform experiments on large datasets and large networks. We perform experiments on MNIST and CIFAR10, which should be adequate for basic research, and very large datasets and models are dispensable to be required in a paper of basic study of deep learning.

---

> ### Author Response · Authors · 2021-11-22
> **Large resnet and CIFAR100 results added**
>
> We have updated the draft with new experiments on resnet20 and CIFAR100. There are three methods in our paper to verify the variance principle. We have done experiments on the first and the second method and obtain similar results in such complicated dataset and network structure. Note that we do not use the third method in the resnet20, because this method requires computing Hessian while computation cost is too large.

---

> ### Author Response · Authors · 2021-12-09
> **Transformer and Multi30k results added**
>
> Dear reviewers,
>
> We have updated the draft with new experiments on transformer and Multi30k dataset. We have done experiments on the first and the second method and obtain similar results in the new experimental setup. Up to now, our experiments are conducted over several representative datasets, i.e., MNIST, CIFAR-10, CIFAR-100 and Multi30k, and network structures, i.e., fully-connected neural networks, VGG-9. ResNet-20 and transformer, thus our conclusion is a rather general result. Since we cannot update the draft on openreview, we post the new version on the anonymous website (details can be found on https://www.dropbox.com/s/5kd7q7wevi984yl/Why_dropout_finds_flatter_minima.pdf?dl=0). Based on a large number of more complex experiments, we expect reviewers to raise the score appropriately.
>
> Sincerely yours,
>
> Authors.

---

### Official Review · Reviewer_HpHx · 2021-11-02

**Correctness:** 3
**Technical Novelty And Significance:** 2
**Empirical Novelty And Significance:** 2
**Recommendation:** 5
**Confidence:** 3

**Main Review:**

In general, the paper is interesting; however, the novelty and contribution are limited. Most of the tools are from the existing work, and neither insights nor empirical results are strong enough to make this paper stands out.

The paper may miss some important discussions regarding the relationship between dropout, regularization, and generalization bounds, as discussed in [1, 2, 3]. Adding the discussion and extra experiments would increase the value of the current draft.

The paper divides the training process into two phases, i.e., fast convergence and exploration phase. However, this phase division is relatively rough and has no support (authors are encouraged to cite some prior works to support their justification).

The computation method for H on neural networks has never been introduced in the main text, making the current empirical results questionable. The computation cost of $H_i$ remains unclear; this comment also applies to the quality of hessian estimation, which may significantly impact the final numerical observation.

It is a bit hard to identify the (inverse) relation between variance and the flatness in Figure 2 and Figure 3a, unlike the clear pattern in Figure 3b. It would be great if the authors could polish the figures and/or explain the current observation.

## Reference
1. Dropout Training, Data-dependent Regularization, and Generalization Bounds
2. On Convergence and Generalization of Dropout Training
3. A PAC-Bayesian Tutorial with A Dropout Bound

**Summary Of The Paper:**

The paper uses the tools from different prior works to study the flatness brought by dropout. The paper shows that the noise induced by the dropout has a similar structured introduced by SGD that leads the training to find flatter minima.

**Summary Of The Review:**

In general the paper is interesting; however, the novelty and contribution is limited. Most of tools are from the existing work, and neither insights nor empirical results are strong enough to make this paper stands out.
Adding the discussion related to the generalization bound (as pointed in the provided literature) and extra experiments would increase the value of the current draft.

---

> ### Author Response · Authors · 2021-11-16
> **Response to Reviewer HpHx**
>
> $\textbf{Point 1:}$
> Most of the tools are from the existing work.
>
> $\textbf{Reply:}$
>
> Dear reviewer, we would like to argue that this should not be a concern. In science, using existing tools or methods should not be a factor to judge the novelty of a result. Obviously, many important works, not limited to the deep learning, use existing tools to discover important phenomena and no one would focus on their tools rather than the results.
>
> $\textbf{Point 2:}$
> Neither insights nor empirical results are strong enough to make this paper stands out.
>
> $\textbf{Reply:}$
>
> Dropout is a very common trick, without a satisfactory framework to understand why it improves generalization. Our work starts from a perspective which is new in dropout study and explores an essential reason for the good generalization of dropout through flatness. This provides a direction for the further research work.
>
> $\textbf{Point 3:}$
> The paper may miss some important discussions regarding the relationship between dropout, regularization, and generalization bounds, as discussed in [1, 2, 3].
>
> $\textbf{Reply:}$
>
> We would like to thank Reviewer’s helpful reference.
> We have added several literatures that study the relationship between dropout, regularization, and generalization bounds in detail, including the three papers mentioned by reviewer, i.e.,
>
> "Many works aim to find an explicit regularization form of dropout. Wager et al. (2013) studies the explicit form of dropout on linear regression and logistic problem, but for studying non-linear neural network, it is still unclear how to characterize the effect of dropout by an explicit regularization term. McAllester (2013) presents PAC-Bayesian bounds, and Wan et al. (2013), Mou et al. (2018) derives Rademacher generalization bounds. These results show that the reduction of complexity brought by dropout is $O(p)$, where $p$ is the probability of keeping an element in dropout. Mianjy\& Arora (2020) show that dropout training with logistic loss achieves $\epsilon$-suboptimality in test error in $O(1/\epsilon)$ iterations.  All of the above works need specific settings, such as norm assumptions and logistic loss, and they only give a rough estimate of the generalization error bound, which usually consider the worst case. However, it is not clear what is the characteristic of the dropout training process and how to bridge the training with the generalization. In this work, we show that dropout noise has a special structure, which closely relates with the loss landscape. The structure of the effective noise induced by the dropout may be a key reason why dropout can find solutions with better generalization."
>
> $\textbf{Point 4:}$
> The paper divides the training process into two phases, i.e., fast convergence and exploration phase. However, this phase division is relatively rough and has no support.
>
> $\textbf{Reply:}$
>
> The training process of neural networks are usually divided into two phases, which are often observed in experiments (Shwartz-Ziv \& Tishby, 2017). Feng \& Tu (2021) use the second training phase to test the relation between flatness and noise variance on SGD. In this work, we follow their experimental scheme to show the similarity between dropout and SGD.
> We add the following introduction of two training phases:
>
> "The training process of neural networks are usually divided into two phases, fast convergence and exploration phase (Shwartz-Ziv \& Tishby, 2017). Feng \& Tu (2021)'s work focuses on the behavior of networks in the exploration phase. In this work, we follow the experimental scheme in Feng \& Tu (2021) to show the similarity between dropout and SGD."
>
> $\textbf{Point 5:}$
> The computation method for H on neural networks has never been introduced in the main text, making the current empirical results questionable.
>
> $\textbf{Reply:}$
>
> We add the following details of Hessian matrix:
>
> “Hessian matrix is the matrix obtained by the second derivative of the loss function of the neural network with respect to the parameter vector of neural network. Here, the parameter vector is a vector consisting of all the parameters of network.”

---

> > ### Comment · Reviewer_HpHx · 2021-11-17
> > **clarify**
> >
> > Thanks for the responses.
> >
> > I would encourage authors to clarify the following points, e.g.,
> > * the explicit method used to calculate the Hessian for neural network, e.g., Hessian can be computed through power-iteration by back-propagating the matvec of the Hessian. However, this estimation requires a relative error threshold, which will also impact the quality of the hessian estimation. There also exist other ways to estimate the Hessian, but such information is never given in the submitted draft.
> > * it is hard for me to infer the exact paper solely based on the text like "Feng & Tu (2021)". It would be great to provide a complete reference list.
> >
> > I think this submission has its own value, but I agree with other reviewers that this submission needs to include a more comprehensive empirical study, covering different types of tasks and neural architectures, e.g., transformer, graph neural network. The current form of the submission is slightly below the bar of ICLR.

---

> > > ### Author Response · Authors · 2021-11-22
> > > **Response to Reviewer HpHx**
> > >
> > > $\textbf{Point 1:}$
> > > The explicit method used to calculate the Hessian for neural network, e.g., Hessian can be computed through power-iteration by back-propagating the matvec of the Hessian. However, this estimation requires a relative error threshold, which will also impact the quality of the hessian estimation. There also exist other ways to estimate the Hessian, but such information is never given in the submitted draft.
> > >
> > > $\textbf{Reply:}$
> > >
> > > We use the autograd function in pytorch to calculate the gradient and the Hessian matrix of the neural network parameters. For the calculation of the eigenvalues and eigendirections of the Hessian matrix, we use the numpy.linalg.eigh function to calculate.
> > >
> > > $\textbf{Point  2:}$
> > > it is hard for me to infer the exact paper solely based on the text like "Feng \& Tu (2021)". It would be great to provide a complete reference list.
> > >
> > > $\textbf{Reply:}$
> > >
> > > We list below the full names of articles that we have cited several times in our responses. Based on the reviews of reviewers, the new draft has made more changes, so the changes mentioned in the reply can also be found in the new draft, which may make it more convenient for you to find the article we mentioned in the reply.\\
> > > Shwartz-Ziv \& Tishby (2017): Opening the black box of deep neural networks via information.\\
> > > Feng \& Tu (2021): The inverse variance–flatness relation in stochastic gradient descent is critical for finding flat minima.\\
> > > Zhang et al. (2017): Understanding deep learning requires rethinking generalization.
> > >
> > > $\textbf{Point  3:}$
> > > This submission needs to include a more comprehensive empirical study, covering different types of tasks and neural architectures.
> > >
> > > $\textbf{Reply:}$
> > >
> > > We agree with the reviewer's comments. We have updated the draft with new experiments on resnet20 and CIFAR100. There are three methods in our paper to verify the variance principle. We have done experiments on the first and the second method and obtain similar results in such complicated dataset and network structure. Note that we do not use the third method in the resnet20, because this method requires computing Hessian while computation cost is too large.

---

> > > > ### Comment · Reviewer_HpHx · 2021-11-22
> > > > **Re: Response to Reviewer HpHx**
> > > >
> > > > Thank you for providing new results for training ResNet20 on CIFAR-100.
> > > >
> > > > But I am somehow confused with the sentence presented in the response: `We do not perform the experiments of computing Hessian due to too large computation cost. In such complicated dataset and network structure, we obtain similar results.' Since the authors claim that they *did not perform the experiments due to too large computation cost*, I am wondering why the authors can still obtain similar results?
> > > >
> > > > I am aware of the deadline for the paper revision, but I would still strongly suggest authors to include one different dataset like text/speech, even on a simple neural network, as the current empirical insights in the paper are only valid for images.

---

> > > > > ### Author Response · Authors · 2021-11-23
> > > > > **clarify**
> > > > >
> > > > > We have updated the draft with new experiments on resnet20 and CIFAR100. There are three methods in our paper to verify the variance principle. We have done experiments on the first and the second method and obtain similar results in such complicated dataset and network structure. Note that we do not use the third method in the resnet20, because this method requires computing Hessian while computation cost is  too large.
> > > > >
> > > > > Also thanks for the reviewer's comment on new experiments of text/speech. We are not familiar with such setting, but we would do these experiments soon.

---

> > > > > ### Author Response · Authors · 2021-12-09
> > > > > **Transformer and Multi30k results added**
> > > > >
> > > > > Dear reviewers,
> > > > >
> > > > > We have updated the draft with new experiments on transformer and Multi30k dataset. We have done experiments on the first and the second method and obtain similar results in the new experimental setup. Up to now, our experiments are conducted over several representative datasets, i.e., MNIST, CIFAR-10, CIFAR-100 and Multi30k, and network structures, i.e., fully-connected neural networks, VGG-9. ResNet-20 and transformer, thus our conclusion is a rather general result. Since we cannot update the draft on openreview, we post the new version on the anonymous website (details can be found on https://www.dropbox.com/s/5kd7q7wevi984yl/Why_dropout_finds_flatter_minima.pdf?dl=0). Based on a large number of more complex experiments, we expect reviewers to raise the score appropriately.
> > > > >
> > > > > Sincerely yours,
> > > > >
> > > > > Authors.

---

> ### Author Response · Authors · 2021-11-16
> **Response to Reviewer HpHx**
>
> $\textbf{Point 6:}$
> It is a bit hard to identify the (inverse) relation between variance and the flatness in Figure 2 and Figure 3a, unlike the clear pattern in Figure 3b.
>
> $\textbf{Reply:}$
>
> We would like to further explain the results of our experiments and give the correlation between the results and our conclusions. In Fig.2, for small flatness part, the variance of noise induced by dropout is generally large, which indicates that the noise induced by dropout has larger variance in sharp directions. For large flatness part, as the loss landscape becomes flatter, the linear relationship becomes more obvious, and we can see a clearer asymptotic behavior in the results. Overall, we can observe the negative correlation between the eigenvalues and flatness in Fig.2. In Fig.3, we can observe the positive correlation between the eigenvalue and the projection variance. With the structure shown in the results, the noise induced by dropout can help the network escape from the bad minima. We clarify the experimental results in the section 6.1:
>
> "More detailed, for small flatness part, the variance of noise induced by dropout is generally large, which indicates that the noise induced by dropout has larger variance in sharp directions, for large flatness part, as the loss landscape flatter, the linear relationship more obvious, we can see a clearer asymptotic behavior in the results. Overall, we can observe the negative correlation between the eigenvalues and flatness in Fig.2."
>
> We clarify the experimental results in the section 6.2:
>
> "The positive correlation between the eigenvalue and the projection variance show the structure of the dropout noise, which helps the network escape the bad minima."

---

### Official Review · Reviewer_mVLJ · 2021-11-02

**Correctness:** 3
**Technical Novelty And Significance:** 2
**Empirical Novelty And Significance:** 2
**Recommendation:** 5
**Confidence:** 2

**Main Review:**

I am not quite familiar with this area. But I think the topic is quite interesting.

The main concerns are:

1. It seems most of the claims are not theoretically supported. I tried to find something in the supplementary material but only code is found.
2. It is observed that regularization also helps and find flatter minimizer and there is work to bridge the dropout and reguarliation: Dropout Training as Adaptive Regularization. More discussion is encouraged.


**Summary Of The Paper:**

This paper tries to study the effect of dropout by studying the trajectory of optimization via dropout. It is claimed that dropout can help find flatter minimizer.

**Summary Of The Review:**

See main review.

---

> ### Author Response · Authors · 2021-11-16
> **Response to Reviewer mVLJ**
>
> $\textbf{Point 1:}$
> It seems most of the claims are not theoretically supported.
>
> $\textbf{Reply:}$
>
> Dear reviewer, we would like to argue that this should not be a concern.
> Although theoretical analysis is an important part of deep learning, theoretical analysis should not be a mandatory choice for the basic research. This paper is an empirical study on the basic research of deep learning, while the theoretical part is left for further study. Many important papers have obtained important results only through experiments, which has a significant impact on the research of deep learning, for example, the best paper of ICLR: Zhang et al. (2017), Understanding deep learning requires rethinking generalization.
>
> $\textbf{Point 2:}$
> It is observed that regularization also helps and find flatter minimizer and there is work to bridge the dropout and reguarliation.
>
> $\textbf{Reply:}$
>
> We would like to thank Reviewer’s helpful reference.
> We add several literatures that study the relation between regularization and dropout in detail, including, "Dropout Training as Adaptive Regularization." , as mentioned by reviewer, i.e.,
>
> "Many works aim to find an explicit regularization form of dropout. Wager et al. (2013) studies the explicit form of dropout on linear regression and logistic problem, but for studying non-linear neural network, it is still unclear how to characterize the effect of dropout by an explicit regularization term. McAllester (2013) presents PAC-Bayesian bounds, and Wan et al. (2013), Mou et al. (2018) derives Rademacher generalization bounds. These results show that the reduction of complexity brought by dropout is $O(p)$, where $p$ is the probability of keeping an element in dropout. Mianjy\& Arora (2020) show that dropout training with logistic loss achieves $\epsilon$-suboptimality in test error in $O(1/\epsilon)$ iterations.  All of the above works need specific settings, such as norm assumptions and logistic loss, and they only give a rough estimate of the generalization error bound, which usually consider the worst case. However, it is not clear what is the characteristic of the dropout training process and how to bridge the training with the generalization. In this work, we show that dropout noise has a special structure, which closely relates with the loss landscape. The structure of the effective noise induced by the dropout may be a key reason why dropout can find solutions with better generalization."

---

> ### Comment · Reviewer_mVLJ · 2021-11-16
> **Thanks for the reply**
>
> Actually, I find that the experimental setting in this paper is very simple, the experiments are only conducted on simple datsets such as MNIST and CIFART 1o and a few architectures, that is why I think this paper should at least have some theoretical support because it is apparently not a comprehensive experimental paper.

---

> > ### Author Response · Authors · 2021-11-22
> > **Large resnet and CIFAR100 results added**
> >
> > We agree with the reviewer's comments. We have updated the draft with new experiments on resnet20 and CIFAR100. There are three methods in our paper to verify the variance principle. We have done experiments on the first and the second method and obtain similar results in such complicated dataset and network structure. Note that we do not use the third method in the resnet20, because this method requires computing Hessian while computation cost is too large.

---

> > ### Author Response · Authors · 2021-12-09
> > **Transformer and Multi30k results added**
> >
> > Dear reviewers,
> >
> > We have updated the draft with new experiments on transformer and Multi30k dataset.  We have done experiments on the first and the second method and obtain similar results in the new experimental setup. Up to now, our experiments are conducted over several representative datasets, i.e., MNIST, CIFAR-10, CIFAR-100 and Multi30k, and network structures, i.e., fully-connected neural networks, VGG-9. ResNet-20 and transformer, thus our conclusion is a rather general result. Since we cannot update the draft on openreview, we post the new version on the anonymous website (details can be found on https://www.dropbox.com/s/5kd7q7wevi984yl/Why_dropout_finds_flatter_minima.pdf?dl=0).  Based on a large number of more complex experiments, we expect reviewers to raise the score appropriately.
> >
> > Sincerely yours,
> >
> > Authors.

---

### Author Response · Authors · 2021-11-16
**Response to all Reviewers**

Dear reviewers,

We thank the reviewers for your thoughtful and insightful comments.  We have addressed every comment, and believe that, taken together, the reviewers' comments have improved the manuscript significantly. To address reviewers' common concerns, we have refined the definition, added experimental setups, and improved the writings. We hope that the revised manuscript now satisfies the reviewers' requirements.

Sincerely yours,

Authors.

---

### Decision · Program_Chairs · 2022-01-20

**Decision:**

Reject

**Comment:**

The authors make an experimental case that dropout aids generalization by
promoting "flatter minima".

The reviewers felt that the work reported in this paper makes a useful step
forward on a question of central interest.  The consensus view was that the
total weight of evidence presented was not sufficient for publication in
ICLR.  The paper could be strengthened was more extensive and varied experiments
and/or theoretical analysis.